# Differences in the Structure and Diversity of Invertebrate Assemblages Harbored by an Intertidal Ecosystem Engineer between Urban and Non-Urban Shores

Ana Catarina Torres [1,2], Marcos Rubal [1,2,*], Ricardo Costa-Garcia [1,2], Isabel Sousa-Pinto [1,2] and Puri Veiga [1,2]

1   Interdisciplinary Centre of Marine and Environmental Research (CIIMAR) of the University of Porto, Novo Edifício do Terminal de Cruzeiros do Porto de Leixões, Avenida General Norton de Matos, 4450-208 Matosinhos, Portugal; a_catarina_torres@hotmail.com (A.C.T.); r.costa-garcia@hotmail.com (R.C.-G.); ispinto@ciimar.up.pt (I.S.-P.); puri.sanchez@fc.up.pt (P.V.)
2   Department of Biology, Faculty of Sciences, University of Porto, Rua do Campo Alegre s/n, 4169-007 Porto, Portugal
*   Correspondence: marcos.garcia@fc.up.pt

**Abstract:** Nowadays, coastal urbanization is one of the most serious and prevalent pressures on marine ecosystems, impacting their biodiversity. The objective of this study was to explore differences in attributes and biodiversity associated with an intertidal ecosystem engineer, the mussel *Mytilus galloprovincialis* Lamarck, 1819 between urban and non-urban shores. For this, mussel attributes and their associated macrofauna were compared between urban and non-urban rocky shores in the north of Portugal. Results showed that the largest sized mussels were more frequent on urban shores, whereas the smallest size class was only present in non-urban shores. Regarding macrofauna associated with mussels, the number of taxa was significantly higher on non-urban shores. Moreover, the structure of the macrobenhic assemblages was significantly different between urban and non-urban shores. Most important taxa responsible for differences were more abundant on non-urban shores except for *Nucella lapillus*, *Idotea pelagica* and Oligochaeta that were more abundant on urban shores. Therefore, our results showed that the mussel size frequency and the structure of the associated macrobenthic assemblages changed in urban shores. Considering the relevance of mussel beds for biodiversity and human well-being, our results indicate the need of adopting proper management plans to minimize these effects on urban intertidal ecosystems.

**Keywords:** ecosystem engineer; *Mytilus galloprovincialis*; urbanization; intertidal; rocky shores; mussel attributes; macrobenthic assemblages

## 1. Introduction

Organisms that through their behavior and/or morphology can directly or indirectly control the resources available to other species are considered ecosystem engineers [1,2]. These organisms can modulate, maintain or create habitats, increasing heterogeneity and thus, biodiversity [3]. Bivalve mollusks that form aggregates and increase substrate complexity are considered ecosystem engineers [4–6]. Moreover, their filter feeding activity improves the benthic-pelagic coupling and thus, intensifies the input of food [7]. Bivalve shells can also serve as secondary substratum for many sessile or sedentary organisms, whereas mobile organisms live in the spaces among them [4,5]. Bivalves are particularly relevant as ecosystem engineers on intertidal habitats because they are capable of ameliorating the high environmental stress during low tide (e.g., thermal stress, desiccation, heavy rains) or the abiotic and biotic pressures during high tide (e.g., wave action, predation) [3]. Mussel beds are one of the most frequent bivalve aggregations on intertidal habitats, harboring diverse assemblages of invertebrates e.g., [6,8–10].

The Mediterranean mussel *Mytilus galloprovincialis* Lamarck, 1819 is an intertidal filter-feeding bivalve, widely distributed on the Atlantic rocky shores of the Iberian Peninsula,

and is one of the most abundant species at intertidal Portuguese rocky shores [11,12]. Furthermore, this species also plays an important role in intertidal food chains [13] and offers valuable ecosystem services, such as food, coastal protection or elimination of pollutants [14–16]. Moreover, *M. galloprovincialis* has a high commercial value because it is a popular shellfish in the human diet that is extensively explored in several European countries [17]. Moreover, it is the main European bivalve species produced in aquaculture [18]. However, mussel beds, particularly in the intertidal, are subjected to various anthropic disturbances, which may have a great impact on the functioning and stability of their aggregates [19,20]. One of these disturbances is increasing urbanization, which is one of the most widespread and growing threats to coastal ecosystems [21,22]. Coastal urbanization is associated with a higher population density; areas up to 100 km from the coastline harbor three times more population than the global average [23] and predictions point out that in the coming decades, 75% of the population will be concentrated in coastal localities [24]. This phenomenon emerges due to the facilitation of activities such as fishing, industry, tourism and transportation, among others in coastal areas [21].

Despite the benefits of these activities to humankind, they are a relevant source of disturbances, such as chemical contamination of water due to domestic or industrial sewage, trampling, harvesting and the introduction of exotic species and/or habitat fragmentation due to construction of artificial structures (e.g., sea walls) that can cause ecological impacts [22,23,25]. Therefore, it is imperative to study the effects of urbanization, particularly in the intertidal, because it is a very accessible area to human activities [19]. Many studies have explored the different impacts associated with urbanization on intertidal habitats such as trampling or artificial structures [22,26–28]. The effect of urbanization on mussel beds has also received some attention, mainly focused on harvesting and trampling e.g., [29–32] but these studies were focused on mussel populations. However, studies focused on the urbanization effect on assemblages associated with mussels are still scarce. Some studies explored the effect of different stressors, mainly invasions and pollution, commonly associated with coastal urbanization. For example, ref. [33] compared the fauna associated in two species of mussels, one native (*Mytilus galloprovincialis*) and one invasive (*Limnoperna securis*) and [34] compared the diversity harbored by a native mussel (*Perna canaliculus*) and an invading ascidian (*Pyura doppelgangera*). Other studies have dealt with the impacts of pollution, such as [35] which investigated the spatial and temporal structure of the fauna associated with *M. galloprovincialis* in the Thermaikos Gulf (northern Aegean Sea) and [36] where the diversity associated with *M. galloprovincialis* inside and outside ports was monitored. However, on urban shores different stressors act simultaneously and thus, may result in a complex response of assemblages that cannot be inferred from studies considering the effect of isolated stressors [37].

As ecosystem engineers increase biodiversity locally, they are considered valuable conservational targets [3,38]. However, to understand how environmental disturbances modify the diversity of assemblages harbored by ecosystem engineers, empirical data are still needed. The main goal of this study is to explore differences in mussel's attributes and its associated macrobenthic assemblage among urban and non-urban shores. For this purpose, differences in density, size frequency and condition index of mussels and the abundance, taxa richness, Shannon index and the structure of macrofaunal assemblages associated with mussel beds were explored on urban and non-urban shores in the north Portuguese coast.

## 2. Materials and Methods

### 2.1. Study Area, Sampling and Sample Procedure

This study was done in April 2017 at four rocky shores in the north coast of Portugal with different degrees of urbanization. In order to define the degree of urbanization in the studied shores, we used population density as a proxy because a higher population density implies a higher degree of trampling, harvesting, more artificial structures and a high input of domestic and industrial sewage e.g., [23,31,39–41]. Two rocky shores were selected in

the metropolitan area of Porto with more than 2800 residents/km$^2$: Leça (41°12′16.49″ N; 8°42′57.16″ W) and Cabo do Mundo (41°13′30.96″ N; 8°43′3.29″ W), considered as urban shores, whereas two other rocky shores: Moledo (41°50′28.81″ N; 8°52′32.30″ W) and Vila Praia de Âncora (41°49′25.93″ N; 8°52′27.42″ W), placed in areas with a population density lower than 130 residents/km$^2$, were considered as non-urban. A previous study [42] also found that concentrations of heavy metals and nutrients were higher in urban shores (Leça and Cabo do Mundo) than in those considered as non-urban (Vila Praia de Âncora and Moledo).

The Portuguese north coast is largely straight and thus all the studied shores have the same orientation and similar environmental conditions. The studied area is characterized by a semidiurnal tidal regime, with the largest spring tides about 4.0 m and most common waves from the west and northwest direction, showing a range of variation from 1.5 to 7 m [43]. The studied rocky shores are characterized by typically granitic substrate and the area presents a fragmented coastal landscape due to the presence of estuaries and varies from soft to hard substrata. Moreover, there is a seasonal upwelling during the spring and summer months, which provides nutrients for organisms [44].

At each shore, two sites separated about 10 m apart were randomly selected, and at each site, five quadrats (10 × 10 cm) in *M. galloprovincialis* beds at mid-tide level were collected to explore potential impacts of urbanization by comparing the attributes of mussels and their associated macrofauna between urban and non-urban shores. For each replicate, samples were collected by scrapping all the quadrat area. All samples were placed in labelled plastic bags. At the laboratory, all samples were frozen (−20 °C) until their processing. Then, each sample was washed through a sieve of 0.5 mm, in order to separate the macrofauna from the mussels, and all mussels in each sample were counted (density). The residue on the sieve was stored in formaldehyde (4%) stained with Rose of Bengal, until sorting and identification of invertebrates to the lowest possible taxonomic level, usually species. From each replicate, 20 mussels were randomly separated to measure their shell length and each individual was assigned to a specific size class: Class 1: <5 mm, Class 2: 5–15 mm, Class 3: 15–25 mm, Class 4: 25–35 mm, Class 5: 35–45 mm and Class 6: >45 mm. Moreover, 10 mussels per replicate were used to calculate the condition index (i.e., the ratio between dry weight of soft tissue and dry weight of the shell) by drying the mussels at 60 °C for 48 h.

### 2.2. Data Analysis

To explore differences between urban and non-urban shores on mussel attributes (i.e., density and condition index) and on the number of individuals (N), taxa richness (S) and Shannon index values (H′) of the invertebrate assemblage associated with mussels, analyses of variance (ANOVAs) were done. For mussel density, N, S and H′, a three-way model was considered, including the factors: Condition (Co) fixed, orthogonal with two levels (i.e., urban and non-urban), Shore (Sh) random with two levels, nested in Co and Site (Si) random, with two levels, nested in Co and Sh, with 5 replicates. For condition index, a four-way model was considered including the same factors described above and Quadrat (Qu) as an additional random factor nested in Co, Sh and Si with 5 levels and 10 replicates. Cochran's test was done to check the homogeneity of variances previously by ANOVA tests. Data were log-transformed to remove the heterogeneity of variances, when necessary. When this was not possible, untransformed data were analyzed and the results were considered robust if significant at $p < 0.01$ [45].

In order to explore differences in size structure of mussels between urban and non-urban shores, their size-frequency was compared by means of Kolmogorov–Smirnov tests (KS).

Permutational multivariate analysis of variance (PERMANOVA, [46]), based on the Bray–Curtis untransformed dissimilarity matrix, was used to analyze the multivariate assemblage data. The model for this analysis was the same as previously described for the three-way ANOVA, using a maximum of 999 permutations in the reduced model with a

defined level of significance, a priori, at $p < 0.05$. When the number of unique permutations for a factor was lower than 30 (or close to 30), Monte Carlo p-values were considered [47].

Multivariate patterns were illustrated by non-metric multidimensional scaling (nMDS) ordination of sampled sites. The PERMDISP procedure was done to test whether differences between urban and non-urban conditions were due to different multivariate dispersion in the location of centroids [48]. Moreover, the SIMPER procedure [49] was used to determine the percentage of contribution ($\delta\%$) of each taxon to the Bray–Curtis dissimilarity between conditions. A taxon was considered important if its contribution to total percentage dissimilarity was $\geq 3\%$. The ratio $\delta/SD(\delta)$ was used to quantify the consistency of the contribution of taxa to the average dissimilarity in all pair-wise comparisons of samples between conditions. Values $\geq 1$ indicated a high degree of consistency.

## 3. Results

### 3.1. Mussel' Attributes

No significant differences between conditions were detected for density and condition index of mussels (Tables 1 and 2; Figure 1).

**Table 1.** ANOVA analysis for density of mussels between conditions. ns: not significant.

| Source of Variation | Df | Density | |
|---|---|---|---|
| | | MS | F |
| Condition | 1 | 110.24 | 0.76 |
| Shore | 2 | 145.66 | 1.01 |
| Site | 4 | 144.04 | 1.29 |
| Residual | 32 | 111.59 | |
| Total | 39 | | |
| Cochran's test | | 0.31 (ns) | |
| Transformation | | Sqrt (X + 1) | |

**Table 2.** ANOVA analysis for condition index of mussels between conditions. s: significant.

| Source of Variation | Df | Condition Index | |
|---|---|---|---|
| | | MS | F |
| Condition | 1 | 0.0277 | 70.97 |
| Shore | 1 | 0.0073 | 8.54 |
| Site | 4 | 0.0009 | 0.88 |
| Quadrat | 32 | 0.0010 | 1.64 |
| Co x Sh | 1 | 0.0004 | 0.46 |
| Residual | 360 | 0.0006 | |
| Total | 399 | | |
| Cochran's test | | 0.8620 (s) | |
| Transformation | | None | |

Size–frequency distribution of mussels was significantly different between conditions (KS test, Dmax = 12.75, $p = 0.004$; Figure 2). On the non-urban shores, mussels belonging to size class 15–25 mm dominated in relation to other size categories, whereas on urban shores a clear dominant size class was not found (Figure 2). Moreover, the smallest size class (<5 mm) was only present in non-urban shores, whereas the largest size class (>45 mm) was only present in urban shores where the three largest size classes (25–35 mm, 35–45 mm and >45 mm) were more frequent than in non-urban shores (Figure 2).

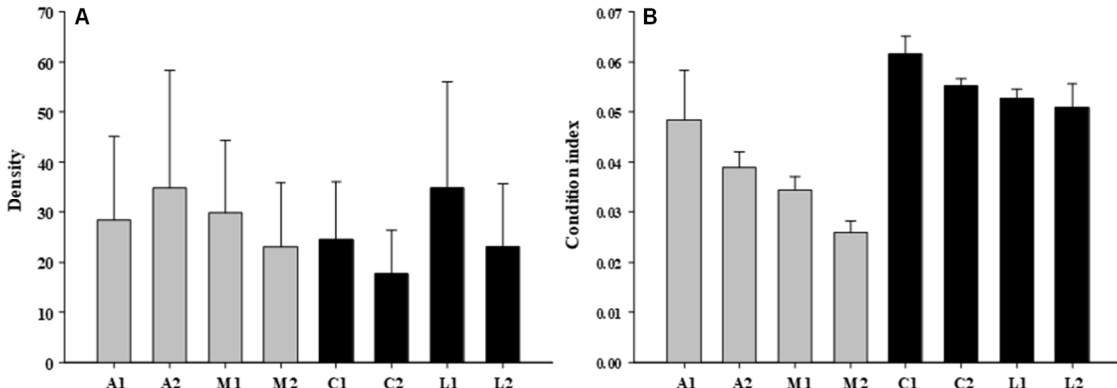

**Figure 1.** Mean values (±SE) of mussel attributes. Density: Sqrt(X+1) transformed (**A**) and Condition index (**B**). Grey: non-urban shores (A1/A2: Âncora site 1 and 2, M1/M2: Moledo site 1 and 2); Black: urban shores (C1/C2: Cabo do Mundo site 1 and 2, L1/L2: Leça site 1 and 2).

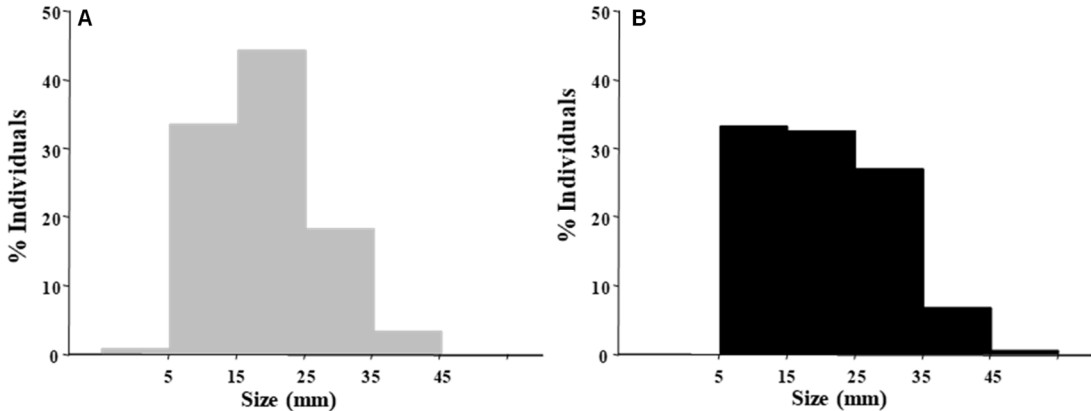

**Figure 2.** Size-frequency of mussels in the two conditions. Grey: non-urban shores (**A**); Black: urban shores (**B**).

### 3.2. Invertebrate Assemblages

A total of 4798 individuals (2650 in non-urban shores and 2148 in urban shores) belonging to 58 different taxa (53 in non-urban shores and 40 in urban shores) were identified. ANOVA indicated significant differences between conditions for S, with significantly higher values on non-urban shores (Table 3; Figure 3). However, no significant differences between conditions were found for N and H' (Table 3, Figure 3).

**Table 3.** Summary of ANOVAs for the number of taxa (S), number of individuals (N) and Shannon index (H') of faunal assemblage associated with *M. galloprovincialis*. *: $p < 0.05$; **: $p < 0.01$; ns: not significant.

| Source of Variation | Df | S | | N | | H' | |
|---|---|---|---|---|---|---|---|
| | | MS | F | MS | F | MS | F |
| Condition | 1 | 0.3231 | **36.83 *** | 12.4339 | 10.39 | 0.2738 | 1.39 |
| Shore | 2 | 0.0088 | 0.21 | 1.1972 | 0.05 | 0.1966 | 0.55 |
| Site | 4 | 0.2343 | 1.27 | 23.0901 | 1.15 | 0.3550 | 4.17 ** |
| Residual | 32 | 0.1845 | | 20.1177 | | 0.0852 | |
| Total | 39 | | | | | | |
| Cochran's test | | 0.3708 (ns) | | 0.3130 (ns) | | 0.2376 (ns) | |
| Transformation | | Ln (X) | | Sqrt (X + 1) | | None | |

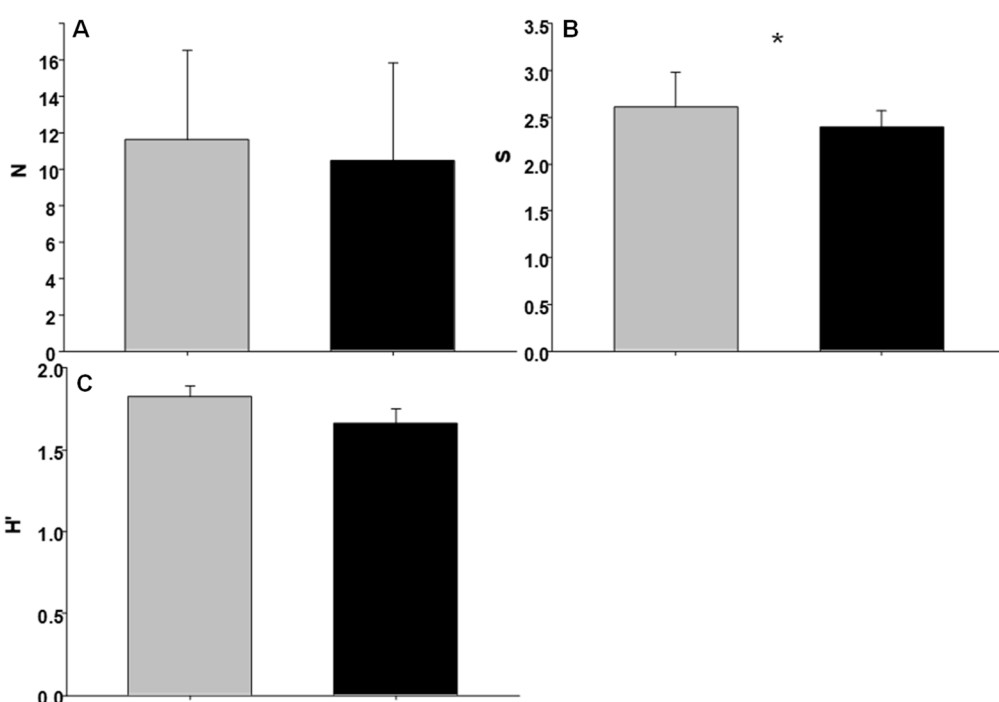

**Figure 3.** Mean values (±SE) of the number of individuals Sqrt (X + 1) transformed (**A**), number of taxa Ln (X) transformed (**B**) and Shannon index (**C**) of macrobenthic assemblages associated with mussels. *: indicate significant differences between conditions ($p < 0.05$). Grey: non-urban shores; Black: urban shores.

Results of PERMANOVA analysis of the structure of the whole assemblage indicated significant differences between conditions (Table 4). The nMDS ordination showed a clear separation between conditions (Figure 4). The PERMDISP (F = 2.5487, $p = 0.149$) indicated that dispersion of samples did not provide a significant contribution to differences detected by PERMANOVA.

**Table 4.** Summary of PERMANOVAs for total assemblage. *: $p$ (MC) < 0.05.

| Source of Variation | Df | Total Assemblage | | |
|---|---|---|---|---|
| | | MS | Pseudo-F | Unique Perms |
| Condition | 1 | 13928 | 3.9564 * | 3 |
| Shore | 2 | 3520.4 | 1.0776 | 296 |
| Site | 4 | 3266.9 | 1.6877 * | 998 |
| Residual | 32 | 1935.7 | | |
| Total | 39 | | | |

The SIMPER analysis identified thirteen taxa as being mainly responsible for differences between conditions. For the total dissimilarity, the percentage of contribution of Nematoda, *Nucella lapillus* (Linnaeus, 1758), *Idotea pelagica* Leach, 1816 *Hyale* spp., *Lasaea rubra* (Montagu, 1803), Oligochaeta, *Brachystomia scalaris* (MacGillivray, 1843) and *Jaera praehirsuta* Forsman, 1949 accounted for almost 80% and the individual contribution of each one was ≥4% (Table 5). All the species with the exception of *N. lapillus*, *I. pelagica* and Oligochaeta were more abundant on non-urban shores (Table 5).

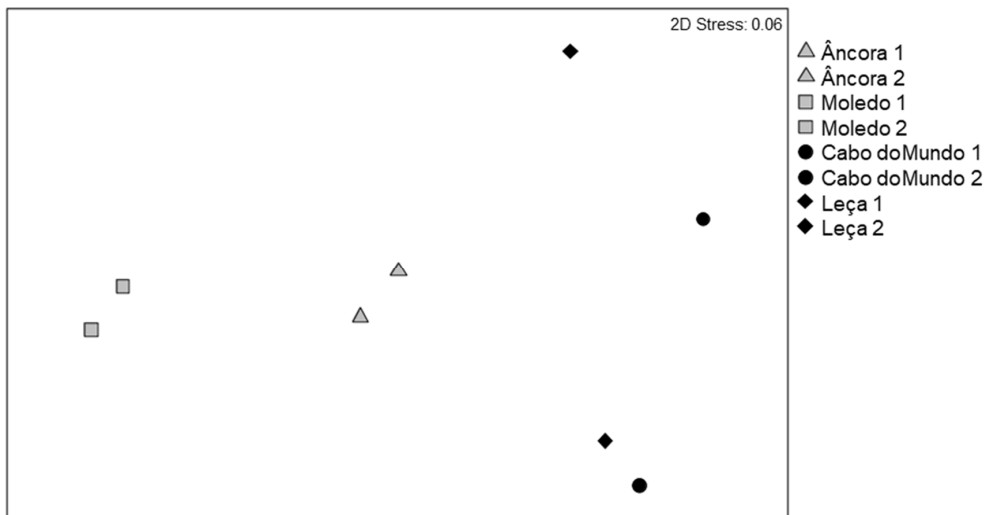

**Figure 4.** nMDS plots for sites at each urban (black) and non-urban (grey) sites.

**Table 5.** Contribution (δ) of individual taxa from faunal assemblages of mussels to the average Bray-Curtis dissimilarity between urban and non-urban shores. Taxa in bold were more abundant in urban shores.

| Taxon | Average Abundance | | $\delta i$ | $\delta i\%$ | $\delta i$/SD ($\delta i$) |
|---|---|---|---|---|---|
| | **Non-Urban** | **Urban** | | | |
| **Nematoda** | **48.00** | **16.15** | 17.29 | 23.80 | 1.34 |
| ***Nucella lapillus*** | 8.75 | 36.20 | 13.21 | 18.18 | 1.15 |
| ***Idotea pelagica*** | 4.85 | 16.70 | 7.71 | 10.61 | 1.02 |
| *Hyale* spp. | 11.95 | 3.45 | 4.65 | 6.40 | 1.03 |
| *Lasaea rubra* | 11.75 | 6.65 | 4.52 | 6.22 | 0.84 |
| **Oligochaeta** | 1.95 | 12.15 | 4.40 | 6.06 | 0.72 |
| *Brachystomia scalaris* | 9.30 | 4.05 | 3.62 | 4.98 | 1.02 |
| *Jaera praehirsuta* | 6.00 | 3.80 | 2.99 | 4.12 | 0.86 |
| *Steromphala umbilicalis* | 4.40 | 2.40 | 2.31 | 3.18 | 0.91 |
| *Sabellaria alveolata* | 6.50 | 0.05 | 1.66 | 2.28 | 0.34 |
| *Apohyale prevostii* | 2.25 | 1.10 | 1.45 | 2.00 | 0.75 |
| *Syllis pulvinata* | 2.15 | 0.35 | 1.22 | 1.68 | 0.36 |
| *Patella depressa* | 2.65 | 0.75 | 1.08 | 1.49 | 0.80 |

## 4. Discussion

Increasing urbanization has become one of the most serious problems of our time for coastal ecosystems [21,23]. Urbanization increases stress sources such as contamination, trampling, harvesting and the introduction of exotic species, among others e.g., [50]. Since *M. galloprovincialis* is able to tolerate disturbances, it is present on many impacted urban shores which offer habitat and resources for other invertebrate species e.g., [35,36]. Therefore, its biomonitoring can provide valuable information about the urbanization impact on intertidal biodiversity on rocky shores by assessing the effects on mussel attributes and its associated macrofauna.

In this study, mussel density and condition index did not show significant differences between urban and non-urban shores. For condition index, ref. [32] found equivalent results in the same study area. However, ref. [32] found a significant decrease in the density of mussels on urban shores. Similarly, a reduction in the abundance of ecosystem engineers was also found in the same study area for native canopy macroalgae [51]. However, a recent study [12] about the spatial and temporal variability of *M. galloprovincialis* along the north Portuguese shore showed that differences in its abundance among shores was dependent on sampling dates. Therefore, the inconsistency between results of [32] and those of our

study may be the result of different sampling dates and natural variability in the abundance of *M. galloprovincialis* on the studied area rather than the effect of urbanization.

Regarding size frequencies of *M. galloprovincialis* in our study, urban shores displayed a higher frequency of larger mussels than non-urban shores, whereas the smallest mussel size was only present in non-urban shores. Similarly, ref. [32] also found the same pattern in the north Portuguese coast, suggesting a low recruitment rate on urban shores. Another plausible reason for the largest size of mussels in urban shores could be the wastewater input from domestic sewage, frequent in urban areas, that could increase the availability and quality of food and benefit mussels as filter-feeding organisms [52,53]. The higher nutrient content in the studied urban shores reported by [42] suggest a higher input of urban sewage, supporting the previous explanation. Similar results were also found in mussels near fish farms whose organic supply makes mussels reach a greater size than mussels far from the farms [54,55]. In contrast, ref. [56] showed that *M. edulis* in urban sites exposed to chemical pollution presented smaller sizes than in reference sites. Despite the tolerance of mussels to pollution, mixtures of chemical pollution such as metals and PAHs may reduce their fitness and, thus, their size [57].

Regarding the macrofauna associated with mussels, taxa richness and the structure of the total macrobenthic assemblage were significantly different between urban and non-urban shores. Taxa richness showed lower values on urban shores. Previous studies on the same study area, assessing urbanization effects on marine rocky shore assemblages, also found similar results on the structure of tidepool assemblages [42] and on canopy forming macroalga diversity [51]. A recent study comparing the community structure among artificial and natural habitats (eelgrass bed, intertidal flats and subtidal bottom) in an urbanized semi-enclosed coastal sea in Japan, found many sharing species among natural habitats; however, the breakwater showed bit sharing species with natural habitats and the lowest number of species [22].

Moreover, previous studies exploring the effects of pollution on fauna associated with mussels also reported losses of biodiversity and alterations in the structure of benthic assemblages. For example, ref. [58] found that the distribution of polychaete species associated with the mussel *Brachidontes rodriguezii* was related to the gradient of organic matter associated with a sewage outfall. Çinar [36] compared the fauna associated with *M. galloprovincialis* inside and outside a port, that has been exposed to numerous pollution discharges since 1960 and is considered as one of the most polluted environments of the Mediterranean Sea. Their results found that the biomass and the number of individuals reached higher values inside the ports, but similar to our urban shores, the number of species was lower in ports. Thus, it appears that the degree of pollution, commonly associated with urban areas, can deeply affect the distribution, composition and abundance of the species harbored by mussels. Therefore, urbanization may change mussel attributes but also the number of taxa and the structure of invertebrate assemblages associated with mussel beds. Several studies have explored how the attributes of mussels are related to the associated fauna. For example, ref. [35] found no relationship between density of mussels and the abundance and diversity of fauna associated with mussels, as in our study, where patterns of mussel density and those of associated fauna were different. However, ref. [10] analyzing the macroinvertebrate communities associated with *M. galloprovincialis* in different regions of a South African estuary reported a negative relationship between the diversity of fauna and the density of mussels, but they found no relationship between mussel size and species richness. On the other hand, ref. [59] carried out a manipulative study to assess whether mussel size affected their associated fauna. They found that in one of the studied locations, the fauna associated with larger mussels differed significantly from the fauna associated with smaller mussels. Nevertheless, the size did not affect the species' richness, but rather the abundance and proportion of the organisms present. When comparing these results with ours, it can be suggested that differences in the mussel size frequency between urban and non-urban shores may affect the structure of the macrobenthic assemblages, showing lower values of species richness in urban shores, where the

highest mussel size frequencies were found. However, another potential explanation is that the reduction in diversity on urban shores may be due to changes in the identity and sensitivity of species to anthropic disturbances [60]. Similarly, ref. [9] also suggests that the differences in fauna composition associated with *M. galloprovincialis* may be due to changes in the water quality. This explanation could be applied to our study, since the studied urban and non-urban shores have different water quality through nutrient enrichment and heavy metal content [42]. In view of this, some species could be more sensitive to urbanization, whereas others could prefer urbanized locations. When analyzing the species that most contributed to the differences in the structure of the community between urban and non-urban shores, only three taxa were more abundant on urban shores (i.e., *Nucella lapillus*, *Idotea pelagica* and Oligochaeta), while the remaining taxa were more abundant on non-urban shores. Among the latter, Nematoda, *Hyale* spp. and *Lasaea rubra* were the most relevant in shaping differences between urban and non-urban shores.

Oligochaetes are considered biological indicators of pollution being more abundant in polluted locations and commonly associated with organic enrichment [61,62]. Oligochaetes, in our study, were more abundant in urban shores probably as a consequence of the high concentration of heavy metals and nutrient enrichment in urban shores [42]. Nematodes are also considered good indicators of the environmental status, since they encompass both sensitive and tolerant species to pollution [63]. Our results seem to indicate that the nematode species found in our study should be more sensitive to disturbances, as they were more highly abundant in non-urban shores. Moreover, urban shores, due to wastewater input from domestic sewage, could accumulate a lot of dead organic matter and favor omnivorous scavenger animals such as *Idotea pelagica*, [64], explaining its higher abundance in urban shores. In contrast, amphipods are usually considered sensitive to pollution and, therefore, occur in lower numbers in polluted locations [62,65] as we found for *Hyale* spp. in urban shores.

In the case of *Nucella lapillus*, this species is known to be very sensitive to tributyltin (TBT) (biocide and anti-fouling paint) [66], and it is used as a bioindicator of coastal system recovery [67]. However, there is not much available information in relation to other contaminants or disturbances [68]. Moreover, it is also known that in urban shores, there is more recreational activity and people usually catch animals for food or bait. *Nucella lapillus* is one of the species subjected to harvest, so it would also be expected that its abundance is lower in urban sites [69]; however, this was not observed in our study. A possible explanation for its higher abundance in urban shores may be its size, because being a large gastropod, *Nucella* could prefer larger mussels (more frequent on urban shores) as these provide greater interstitial spaces in the mussel aggregates. The same pattern was found for oligochaetes and *Idotea*, which prefer larger mussels [59]. *L. rubra* was found that thrived near a sewage outfall [53], so one would expect higher abundances of this species in urban shores, but this was not found in our study. This may be due to its size. Being a small species, it may prefer habitats with smaller interstitial spaces, such as those provided by smaller mussels, more frequent on non-urban shores. Moreover, *Lasaea* is part of the diet of *N. lapillus* juveniles, that were more abundant in urban shores and thus, predation intensity in urban shores may help to reduce *Lasaea* abundance [70]. Therefore, the differences found in the structure of macrobenthic assemblages between urban and non-urban shores can be linked to changes in the quality of the habitat, through modifications in the mussel size frequency but also in the species' sensitivity or tolerance to anthropic disturbances associated with increased urbanization [23,60].

In conclusion, our study showed significant differences between urban and non-urban shores for the mussel size frequency, as well as for taxa richness and the structure of the total macrobenthic assemblages associated with *M. galloprovincialis*. These results support previous studies on other taxa that suggested changes on the traits of ecosystem engineers and a reduction in biodiversity harbored by urban shores. Therefore, the adoption of proper management plans is needed to prevent and minimize the loss of diversity in urban intertidal ecosystems that may reduce human well-being [71,72].

**Author Contributions:** Conceptualization, P.V. and M.R.; methodology, A.C.T., M.R., R.C.-G. and P.V.; formal analysis, A.C.T. and M.R.; investigation, A.C.T., M.R., R.C.-G. and P.V.; resources, M.R., P.V. and I.S.-P.; writing—original draft preparation, A.C.T.; writing—review and editing, I.S.-P., M.R. and P.V.; supervision, I.S.-P., M.R. and P.V.; project administration, P.V.; funding acquisition, P.V. All authors have read and agreed to the published version of the manuscript.

**Funding:** This research was developed under the Project No. 30181 (PTDC/CTA-AMB/30181/2017), co-financed by COMPETE 2020, Portugal 2020 and the European Union through the ERDF, and by the FCT-Foundation for Science and Technology through national funds within the scope of UIDB/04423/2020 and UIDP/04423/2020 and the PhD scholarship (SFRH/BD/114935/2016) to ACT.

**Data Availability Statement:** Not applicable.

**Acknowledgments:** We are grateful to two referees for all the helpful comments and suggestions, which greatly improved this paper. P Veiga was hired through the Regulamento do Emprego Científico e Tecnológico—RJEC from the Portuguese Foundation for Science and Technology (FCT) program (CEECIND/03893/2018).

**Conflicts of Interest:** The authors declare no conflict of interest. The funders had no role in the design of the study; in the collection, analyses, or interpretation of data; in the writing of the manuscript, or in the decision to publish the results.

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
