# Peer review of "Differences in the Structure and Diversity of Invertebrate Assemblages Harbored by an Intertidal Ecosystem Engineer between Urban and Non-Urban Shores"

_jmse, doi:10.3390/jmse10020242_

Round 1

Reviewer 1 Report

Torres et al. investigate the effects of urbanization on mussel beds and associated macrobenthic assemblages. To achieve their goal, they sampled four rocky shores (two urban and two non-urban) on the north coast of Portugal and assessed differences in mussel density, size-frequency, and condition index of mussels as well as changes in the number of species and individuals of macrobenthic organisms. The authors did not find major differences in density and condition index of mussels; yet significant changes were found in the number of species in the associated macrobenthic fauna.

Overall, I found the manuscript interesting and well-structured. The objectives and main results are presented clearly, and the manuscript is generally well-written (although some errors and typos can be found throughout the manuscript, which should be revised before publication). The authors found and identified a large number of organisms (almost five thousand individuals belonging to 58 species) and the results may provide important information on the effects of urbanization on coastal biodiversity. I, therefore, commend the authors for their work.

My main concern, however, is about the number of sites sampled: There are only two sites sampled in each category (urbanized vs non-urbanized), and I am not sure how they accurately represent urbanization effects. Additionally, even the non-urbanized site seems to be densely populated (lower than 50 000 residents/km2). As the authors have sampled other areas (as showed by previous papers published by the research group and cited in this manuscript), I wonder if it is not possible to include data from previous works to strengthen current conclusions. In my opinion, this would provide further evidence on the influence of urbanization on coastal biodiversity.

Specific comments:

TITLE

It is informative and a good reflection of the content.

 ABSTRACT

The abstract covers the main points of the paper.

 INTRODUCTION

The problem is clearly stated and it is also clear why the manuscript is important.

 METHODS

The methods are described clearly; however, I believe more information could be added. For example, it is important to mention if all samples were collected at the same height of rocky shore. Moreover, please state whether all sampling areas have similar environmental conditions. It is well-known that local conditions such as wave action may strongly influence rocky shore biodiversity and surpass the effects of urbanization. In this context, it would be important to control for the influence of these co-variables.

DATA/RESULTS

As mentioned before, I would like to have more information on the environmental characteristics of each area and how they influence local biodiversity.

DISCUSSION

The results are discussed properly and the authors made clear the importance of their work. Yet, the conclusions are rather vague and beyond the scope of the manuscript. I recommend the authors to focus on their results and bring direct implications of their achievements.

 FIGURES/TABLES

Figures and tables are clear and informative. However, I suggest the authors highlight which group was more abundant in each environment in table 5.

Author Response

We included an acknowledgement section in the revised manuscript.

We are grateful to two referees for all the helpful comments and suggestions, which greatly improved this paper. P Veiga was hired through the Regulamento do Emprego Científico e Tecnológico—RJEC from the Portuguese Foundation for Science and Technology (FCT) program (CEECIND/03893/2018).

 Referee 1

My main concern, however, is about the number of sites sampled: There are only two sites sampled in each category (urbanized vs non-urbanized), and I am not sure how they accurately represent urbanization effects. Additionally, even the non-urbanized site seems to be densely populated (lower than 50 000 residents/km2). As the authors have sampled other areas (as showed by previous papers published by the research group and cited in this manuscript), I wonder if it is not possible to include data from previous works to strengthen current conclusions. In my opinion, this would provide further evidence on the influence of urbanization on coastal biodiversity.

 In order to detect changes on the mussel population and on its associated invertebrate assemblages we applied a hierarchical sampling design increasing the sampling effort at the scales where high variability is found. We considered the great amount of literature that has proved that variability is higher at the smallest spatial scale (i.e. among replicates), also in the study area, and thus, we collected 5 replicates at each sampling site. This is a total of 20 replicates for each condition (urban and no-urban). At the scale of tens of meters, we considered two sites within each shore (i.e. 4 sites for urban and 4 shores for non-urban condition) and finally at the scale of kilometers, where variability is lower, we considered two shores at each condition. Therefore, we think that we have included several relevant spatial scales with a proper replication to explore differences between urban and non-urban shores. To more information about scales of variation in intertidal rocky shores please see the following references:

Fraschetti S, Terlizzi A, Benedetti-Cecchi L (2005) Patterns of distribution of marine assemblages from rocky shores: evidence of relevant scales of variation. Mar Ecol Prog Ser 296:13–29

Benedetti-Cecchi L (2001) Variability in abundance of algae and invertebrates at different spatial scales on rocky sea shores. Mar Ecol Prog Ser 215:79–92

Veiga P, Rubal M, Vieira R, Arenas F, Sousa-Pinto I. 2013. Spatial variability in intertidal macroalgal assemblages on the North Portuguese coast: consistence between species and functional group approaches. Helgol Mar Res 67:191–201.

Regarding to the number of residents per km2 we made a mistake including the decimal values. Correct updated values were included in the corrected version of the manuscript. You will see than urban areas have a population density one order of magnitude higher than non-urban areas.

We have considered similar previous studies in the same areas in the discussion section (references 12, 32, 42, 51). However, two of them considered different target taxa (macroalgae and tidepool assemblages) and the ones that considered mussels have a different design and did not consider the associated invertebrate assemblage. For these reasons we considered these papers in the discussion but we cannot include their data on this manuscript.

Specific comments:

 METHODS

The methods are described clearly; however, I believe more information could be added. For example, it is important to mention if all samples were collected at the same height of rocky shore. Moreover, please state whether all sampling areas have similar environmental conditions. It is well-known that local conditions such as wave action may strongly influence rocky shore biodiversity and surpass the effects of urbanization. In this context, it would be important to control for the influence of these co-variables.

 All samples were collected at the mid-tide level, as mussel beds are restricted to this level in north Portuguese shore. We included this clarification in the manuscript.

The coast along the studied area is largely straight without any significant bay, ría or other relevant geological formation, that could modify wave action on the shore. Therefore, the general description about physical environment, including wave regimen, is valid for all the studied shores. We have clarified this point in the material and methods section.

DATA/RESULTS

As mentioned before, I would like to have more information on the environmental characteristics of each area and how they influence local biodiversity.

 See response to the previous comment.

DISCUSSION

The results are discussed properly and the authors made clear the importance of their work. Yet, the conclusions are rather vague and beyond the scope of the manuscript. I recommend the authors to focus on their results and bring direct implications of their achievements.

 Conclusions were modified following both referee comments

 FIGURES/TABLES

Figures and tables are clear and informative. However, I suggest the authors highlight which group was more abundant in each environment in table 5.

Table 5 was modified following this suggestion

Reviewer 2 Report

Major comments:

These results have a value to support previous studies that showed the effects of urbanization on marine ecology. However, the authors did NOT show how urbanization affected them, which was purposed in this study. Authors have to change their purposes and revise in discussion and conclusions in this manuscript.

Minor comments:

L67: “… intertidal, because is …”

A subject is needed after "because".

L81: “… monitored the.”

This sentence is not completed.

L87: “The main goal of this study is to explore the effects of urbanization on mussel’s attributes and its associated macrobenthic assemblage”

If the authors would like to result in the ecological effects of urbanization, they must clarify the factors of urbanized and unurbanized areas. Or, the authors have to change this purpose.

L94: Readers cannot understand the differences in ecology between urbanized and unurbanized areas. The ecological differences have to be shown in this paragraph in detail.

L118: “-20”

Minus sign must be used.

L126: “condition index” is vague. Because the term of “the ratio between dry weight of soft tissue and dry weight of shell” is clearer than “condition index,” it would be recommended to be used through this manuscript.

Figure 3, caption: “…the number of number individuals…” may be “…the number of number individuals…”

Discussion section: The authors have to explain why you can conclude the effects of urbanization on mussel’s attributes and their associated microbenthic assemblages from the results of this study.

L213: This paragraph states known things but not the values of this study and things what is new in this study.

L331: This conclusion can be led without the results in this study. I believe that a value of this study would be supporting things known by previous studies. Authors have to conclude things led from this study.

L446: The DOI number of this article is not completed.

Author Response

We included an acknowledgement section in the revised manuscript.

We are grateful to two referees for all the helpful comments and suggestions, which greatly improved this paper. P Veiga was hired through the Regulamento do Emprego Científico e Tecnológico—RJEC from the Portuguese Foundation for Science and Technology (FCT) program (CEECIND/03893/2018).

Referee 2

Major comments:

These results have a value to support previous studies that showed the effects of urbanization on marine ecology. However, the authors did NOT show how urbanization affected them, which was purposed in this study. Authors have to change their purposes and revise in discussion and conclusions in this manuscript.

 We have modified the title of the manuscript, abstract, the objective and conclusions of the manuscript to clarify this issue.

Minor comments:

L67: “… intertidal, because is …”

A subject is needed after "because".

 The sentence was corrected

L81: “… monitored the.”

This sentence is not completed.

 We corrected this mistake

L87: “The main goal of this study is to explore the effects of urbanization on mussel’s attributes and its associated macrobenthic assemblage”

If the authors would like to result in the ecological effects of urbanization, they must clarify the factors of urbanized and unurbanized areas. Or, the authors have to change this purpose.

 We understand that we cannot establish a cause effect relationship between urbanization with an observational study. Therefore, we clarified the purpose of this study to correct this mistake.

L94: Readers cannot understand the differences in ecology between urbanized and unurbanized areas. The ecological differences have to be shown in this paragraph in detail.

 We modify this paragraph to clarify the human impacts that are related to urbanization and affect urban shores.

L118: “-20”

Minus sign must be used.

 Correction was done

L126: “condition index” is vague. Because the term of “the ratio between dry weight of soft tissue and dry weight of shell” is clearer than “condition index,” it would be recommended to be used through this manuscript.

The name “condition index” is widespread in the bivalve literature, particularly in aquaculture and or management of natural bivalve resources. In any case, we included a definition of condition index and thus, we think that we must maintain this name.

Figure 3, caption: “…the number of number individuals…” may be “…the number of number individuals…”

 The mistake was corrected

Discussion section: The authors have to explain why you can conclude the effects of urbanization on mussel’s attributes and their associated microbenthic assemblages from the results of this study.

 This issue was corrected (see response to major comments)

L213: This paragraph states known things but not the values of this study and things what is new in this study.

 We agree with the referee but, we included that paragraph as an introduction for the discussion. Results of this study are discussed below.

L331: This conclusion can be led without the results in this study. I believe that a value of this study would be supporting things known by previous studies. Authors have to conclude things led from this study.

 Conclusion were modified following recommendation from both referees

L446: The DOI number of this article is not completed.

This mistake was corrected

Round 2

Reviewer 2 Report

L218: As a manner for scientific journal I believe, the summary of authors results and new findings in this study have to be stated in the first paragraph in Discussion. If the Editor agrees this manner in this journal, I also accept author’s no replying to my comment.

Author Response

Thank very much for your comment.

We have checked the style of recent papers by JMSE and we found many different options for this issue. Therefore, we decided to maintain the current formant, to be consistent with other publications of the group.

Best regards